# A Review of Critical Issues in High-Speed Vat Photopolymerization

**DOI:** 10.3390/polym15122716

**Published:** 2023-06-17

**Authors:** Sandeep Kumar Paral, Ding-Zheng Lin, Yih-Lin Cheng, Shang-Chih Lin, Jeng-Ywan Jeng

**Affiliations:** 1High Speed 3D Printing Research Center, National Taiwan University of Science and Technology, No. 43, Sec. 4, Keelung Rd., Taipei 106, Taiwan; 2Department of Mechanical Engineering, National Taiwan University of Science and Technology, No. 43, Sec. 4, Keelung Rd., Taipei 106, Taiwan; 3Graduate Institute of Biomedical Engineering, National Taiwan University of Science and Technology, No. 43, Sec. 4, Keelung Rd., Taipei 106, Taiwan; 4Academy of Innovative Semiconductor and Sustainable Manufacturing, National Cheng Kung University, No. 1, University Rd., Tainan 701, Taiwan

**Keywords:** high-speed VPP, mass customization, separation force, resin refilling, liquid crystal display (LCD), digital light processing (DLP)

## Abstract

Vat photopolymerization (VPP) is an effective additive manufacturing (AM) process known for its high dimensional accuracy and excellent surface finish. It employs vector scanning and mask projection techniques to cure photopolymer resin at a specific wavelength. Among the mask projection methods, digital light processing (DLP) and liquid crystal display (LCD) VPP have gained significant popularity in various industries. To upgrade DLP and LCC VPP into a high-speed process, increasing both the printing speed and projection area in terms of the volumetric print rate is crucial. However, challenges arise, such as the high separation force between the cured part and the interface and a longer resin refilling time. Additionally, the divergence of the light-emitting diode (LED) makes controlling the irradiance homogeneity of large-sized LCD panels difficult, while low transmission rates of near ultraviolet (NUV) impact the processing time of LCD VPP. Furthermore, limitations in light intensity and fixed pixel ratios of digital micromirror devices (DMDs) constrain the increase in the projection area of DLP VPP. This paper identifies these critical issues and provides detailed reviews of available solutions, aiming to guide future research towards developing a more productive and cost-effective high-speed VPP in terms of the high volumetric print rate.

## 1. Introduction

In the 1980s, the first additive manufacturing (AM), also known as 3D printing, was the stereolithography printing technique to fabricate the prototype by adding the material layer by layer [1,2]. Since then, the expeditious development of AM has gained the attention of engineers and researchers. It has been expanded in industries like automobile, metamaterials, aerospace, ceramic, dentistry, electronics, jewelry, biomedical, and construction [3,4,5,6]. The continuous improvement in designing concepts, mainly known as design for additive manufacturing (DfAM) and manufacturing processes based on AM, is quite impressive and has proven to be a game changer in the era of Industry 4.0 [7]. Depending on the working principle, ASTM F42 has classified the AM technologies into seven main categories i.e., vat photopolymerization, powder bed fusion, binder jetting, material extrusion, material jetting, direct energy deposition, and sheet lamination [8,9].

Vat photopolymerization (VPP) is presently one of the most popular and in demand among AM technologies [10], which can be justified by the statistical result shown in Figure 1. Based on different variants, particularly light source, pattern formation, and control system, VPP can be classified as stereolithography (SLA), two-photon polymerization (TPP), and mask projection VPP. SLA utilizes a layer-by-layer vector scanning approach to create structures, while TPP employs a voxel-by-voxel process to fabricate highly detailed microor nano-3D structures without any mask. An exemplary laser has been focused on the photopolymer resin, guided by a scanning galvanometer in the SLA process (Figure 2a). The laser tracing portion of the resin has been cured by photopolymerization. After completing to cure of an entire cross-section layer in the top-down approach, the build platform moves down a distance equal to the layer thickness, then a resin-filled blade sweeps across the vat to recoat with fresh resin [11]. In the TPP process, the high-intensity ultrashort femtosecond laser is tightly focused onto a spot in a photosensitive resin by a high numerical aperture lens. The presence of photoinitiator molecules initiates the polymerization process, generating localized radicals through two-photon absorption at the center of the focused beam, resulting in the formation of a volume pixel (voxel). The schematic view of the two-laser-beam TPP is shown in Figure 2b, although a single laser can use different photoinitiator chemistries [12,13]. Based on image projection techniques, mask projection VPP can be further classified as liquid crystal display (LCD) and digital light processing (DLP) [14]. Furthermore, the liquid crystal on silicon (LCoS) chip-based dynamic mask projection method has been reported for specifically microstructure printing. The LCoS chip serves dual functions as a mask and shutter, displaying sliced images to modulate the illumination light and enabling control over polymerization time by switching off completely, resulting in a non-reflective surface and no light transmission to the resin bath. [15]. However, this review article focuses on the DLP and LCD VPP due to their high industrial acceptance rate and multipurpose applications. The digital micromirror device (DMD) is the most updated microoptoelectrical mechanical system (MOEMS), containing an orthogonal array of highly reflective aluminum micromirrors, a key component of DLP VPP (Figure 2c). Each micromirror can be controlled individually to direct light onto photosensitive resin which causes the curing at specific patterns [16]. A liquid crystal display has been used as an imaging system in LCD VPP. The light comes from an array of LEDs transmitted through the LCD panel by which a complete cross-section layer has been cured (Figure 2d). According to the slicing image, an electric field has been applied to the LCD panel to change its molecular arrangement and prevent the light from passing through it [17,18,19]. 

The idea of mass customization (MC) is now demanded in the era of Industry 4.0 by which the flexibility and personalization of custom-made products are combined with mass production [21,22]. The concept of MC is not similar to mass production, as shown in Figure 3a. There are four different MC approaches: collaborative, where consumers and manufacturers collaborate to create personalized products; adaptive, offering predefined product variations for different customer segments; cosmetic, focusing on visual aspects to make it personalized; transparent, providing clarity and allowing to track the production process [23]. Of these, collaborative customization is the most challenging for its highest level of product and representation change. In the footwear, eyewear, and jewelry industries, there are a large number of design parameters that vary with customer choice. However, high-speed LCD and DLP VPP have the potential to solve this complexity in a cost-effective manner [24,25,26,27]. There are several advantages of LCD and DLP VPP: (1) high precision, which is possible to achieve by using higher resolution DMD and LCD panel to get fine details in XY plane along with precise control of the stepper motor in Z direction; (2) low surface roughness, the fabricated parts generally have a better surface finish than any other AM processes; (3) cost-effectiveness [28]. It can be seen from Figure 3b,c that the customized 45 pcs shoe midsoles have been produced at the same time in a build platform by using the 32-inch LCD VPP. 

Concerning the present development of LCD and DLP VPP technologies to design a high-speed VPP in terms of the high volumetric print rate, a comprehensive review of the critical issues is necessary to point out the research directions where more focus is required. A comprehensive review of the separation force reduction and modification of the resin refilling methodologies has been carried out as these are among the main constraints to escalating the printing speed and printing area of LCD and DLP VPP. Additionally, the image projection issues of LCD and DLP VPP have been identified, and a constructive review has been made which can assist to recognize the improvement zone. Solving these issues make it feasible to bring out the DLP and LCD VPP as high-speed AM technology to accomplish the demand for mass customization with scalability.

## 2. High-Speed VPP

The development of high-speed VPP can be possible not only to increase the printing speed but also developing a large projection area of LCD and DLP VPP or increment the volumetric print rate that can reduce the production time per unit product. The volumetric print rate is an understandable parameter to judge the high-speed VPP from the industrial concern which can be calculated by multiplying the projection area and printing speed, shown in Table 1. The dynamic mask-based projection (LCD and DLP) VPP helps to overcome the printing time issue compared to other VPP processes. However, it is not well suited to produce functional products from the industrial’s viewpoint. To print a single layer for LCD and DLP VPP, the total time includes the cure time, layer separation time, and resin refill time. The cure time depends on the resin properties and the nature of the projected light source which depends on the applications and cannot change arbitrarily. During the layer separation, the platform needs to move up to separate the cured part from the interface or film, while extra time is also required to fill up the vacuum with uncured photopolymer resin for next-layer printing. This surplus time increases the overall process time which must be minimized to increase the production rate. Thus, it is only suitable for low-volume production. In conventional production, after finishing the mold fabrication, the production time of the injection molding takes only a few seconds for a single part. The development of high-speed vat photopolymerization is one of the most prominent-way to compete with traditional manufacturing processes. Based on the current availability of industrial products (Table 1), it has been observed that generating a high printing speed at large printing areas is still challenging due to the mentioned issues like a high separation force, a longer resin refilling time, and challenges with the optical projection system to maintain high accuracy printing. 

## 3. Separation Force

The separation of the cured layer from the interface or constrained surface, caused by adhesion during photopolymerization, poses a significant challenge in bottom-up VPP processes. It can be seen from Figure 4a that the deformation of the interface (film) occurs due to the separation force. In DLP VPP, a top-down approach is also applied, similar to the platform movement and resin refilling processes of top-down SLA, which is already shown in Figure 2a. The difference is that the DLP projector employs the mask projection to cure the photopolymer resin layer by layer in DLP VPP instead of the laser scanning. However, the bottom-up method is more acceptable and used in industrial applications over the top-down method due to the several advantages:The vat depth is independent of the printing height, and the resin waste is less.Resin refilling happens automatically without any recoating process.The polymerization in the open air is difficult to control due to the oxygen inhibition effect. So, layer solidification is faster in the bottom-up method.The layer thickness is easier to control by the movement of the platform that is not dependent on the resin viscosity.

Different methods that have been proposed to solve the separation force issue still significantly hinder manufacturing capability, especially for a high printing speed and large-area printing. In addition, it is also essential to understand the mechanism (Section 3.1) behind the layer separation which can help to understand the effect of printed part size and geometry, process parameters, interface or film thickness, etc. Pan et al. [38] studied the effect of the printing area-to-perimeter (A/L) ratio by using the polydimethylsiloxane (PDMS) interface. The maximum separation force increases with increasing the ratio of (A/L) though the cross-section area is the same for different geometries, whether the separation force decreases with increasing the hollow portion of the printed geometry by keeping the constant perimeter. The effect of the separation velocity on the separation force has been examined and developed the correlation between them has been by using a 2 mm thick PDMS film [39]. The experimental results show that the separation velocity highly dominates the force-time profile. With the increment of the separation velocity, the separation force increases, and layer separation happens more rapidly. The effect of the geometry and the process parameters are summarized in Table 2. Due to the high separation force, printing defects and print failures, including holes in printed parts, layer separation, and damage in constrained surfaces are supposed to happen [38,40,41,42], as shown in Figure 4b,c.

### 3.1. Separation Mechanisms

At first, it is necessary to understand the separation mechanisms which are primarily concerned with the adhesion characteristic between the rigid body and an elastic solid surface or liquid. The 3D-printed cured layer’s separation from the constraint surface can be elaborated as a laminated composite delamination process like mode-I fracture (due to tensile stress normal to the plane) [38,39,42,45]. Based on this delamination process, the virtual crack closure technique (VCCT) and the cohesive zone model (CZM) are the appropriate methods to model the separation of the cured layers. The VCCT method has been used in the crack delamination process which can only predict crack initiation with pre-existing sharp and neat tip cracks. On the other hand, the CZM method can predict crack initiation and propagation without the help of a pre-existing crack tip and can apply to complex structures subjected to complex loading conditions also [46,47,48,49,50]. Furthermore, the Derjaguin, Muller and Toporov (DMT) and Johnson Kendall Roberts (JKR) separations models have been applied to identify the separation types based on the interface stiffness to consider the cured layer as a rigid punch and the interface as an elastic film of finite thickness [51].

In mode-I crack, the tensile load is applied generally to the crack plane. The generating normal and shear stresses tend to infinitely at the crack tip by using the concept of conventional fracture mechanics, which is unacceptable. The CZM can solve the above issue by dividing the crack length into two different zones, the cohesive and the stress-free crack zone (Figure 5a). The cohesive traction is generated due to the separation of the cohesive surfaces. The separation of the cohesive surfaces leads to crack growth gradually, along with increasing the applied load and continuing up to the complete separation 𝛿_*m*_. For the bilinear traction separation law (TSL) [39], the traction 𝜎 and the separation 𝛿 follow a linear relation from O to point B, and traction reaches the maximum value 𝜎_*m*_ at separation 𝛿_*n*_ from where the damage initiates (Figure 5b). After that, traction follows another linear relation and goes to zero at 𝛿_*m*_. The bilinear TSL can be described (Equations (1) and (2)) by the following distinct parameters *k*, 𝛾.
(1)$$k=\frac{\sigma_{m}}{\delta_{n}}$$
(2)$$\gamma =\frac{\sigma_{m}\delta_{m}}{2}$$

The elasticity (E), the possession ratio (𝜈), and thickness (t) of the interface have been considered to understand the different separation modes like JKR-like and DMT-like separation (Figure 5c). The uniform stress occurs, and the cured sample is detached suddenly in DMT-like separation. However, in JKR-like separation, the interface breaks like a crack propagation, and the separation initiates from periphery to center. The sudden separation of the cured layer causes a higher separation force during DMT separation than JKR. Based on Griffith’s energy balance Kendall et al. [52] developed the maximum pull of force analytically and further modified by Yang [53] by assuming the film thickness is smaller than the radius of the punch (a). Choi et al. [54] considered the shear modulus of the film and the correction factor of the normalized film thickness (t/a) and poisson’s ratio to calculate the separation force. The above analytical solutions are not feasible when the film thickness tends to zero due to the infinitely large separation force created for separation despite finite adhesion energy. In recent research [51,55], the proposed analytical form (Equation (3)) of the separation force is more acceptable than the previous solutions. The separation modes have been identified at first based on the non-dimensional parameter 𝜂 which is defined by the ratio between the elastic deformation of the film or interface (𝛿_*f*_) and the interaction distance of the adhesive force (𝛿_*m*_). The derivation of the separation forces has been conducted depending on the separation modes: DMT like (η<0.6) and JKR like $\left(\eta >30\right).$
(3)$$P_{c}=\left\{\begin{array}{c} \pi \sigma_{m}a^{2},\eta <0.6\\ 4\left(\sqrt{\gamma E}a^{\frac{3}{2}}\right)\sqrt{\frac{2\pi }{3}}\sqrt{1+\frac{4a}{3t}+\frac{4}{3}\frac{a^{3}}{t^{3}}},\eta >30 \end{array}\right.$$

**Figure 5 polymers-15-02716-f005:**
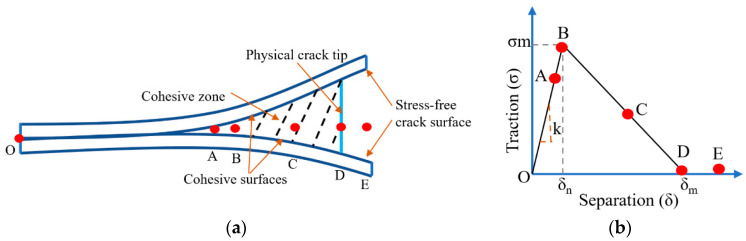
Separation mechanisms for solid–solid interaction. (**a**) Crack propagation based on CZM. (**b**) The traction separation relation of bilinear CZM’s constitutive parameters. (**c**) Detachment of a flat rigid punch from an elastic film. Schematic diagram showing two typical separation modes (DMT like and JKR like) based on 𝜂 [39,55].

The solid–solid separation mechanisms cannot be applied in the presence of the oxygen-aided inhibition layer or ‘dead zone’ where cured object always maintains a distance of h (dead zone thickness) from the interface or film, like solid–liquid interaction (Figure 6a). Based on the Navier–Stokes equation assuming constant viscosity of photopolymer resin, the separation force has been derived for regular solid circular geometry (Equation (4)) and an irregular solid geometry (Equation (5)). The effect of separation velocity V, liquid resin viscosity μ, and the geometry of the printed body have been considered in this study [38]. Of these parameters, the dead zone thickness h and radius R are the most critical parameter to control the separation force of the regular solid geometry. The separation force is inversely proportional to the third power of the dead zone thickness while it is directly proportional to the fourth power printed part’s radius. For irregular solid geometry, the separation force has been influenced by the function of the area-to-perimeter ratio (A/L) and the radius r.
(4)$$P_{c}=\frac{3\pi \mu V}{{2h}^{3}}R^{4}$$
(5)$$P_{c}=\frac{8\mu VA}{h^{3}L}\int_{0}^{2\pi }r\left(\theta \right)d\theta$$

### 3.2. Interface Modifications

The separation of the cured part from the interface is one of the main challenges in bottom-up vat photopolymerization. After the polymerization, the cured part has strongly adhered to the transparent interface or film, causing the object to deform when the pull-up force is applied. To prevent such failure, several types of research have been performed to modify the interface material and design. To make an excellent adhesion of the cured layer with the build plate and or previously cured layer, a higher light exposure time is required to overcome the separation force. However, the over-curing makes an inaccurate model and deteriorates the surface finish. Huang et al. used a hyperelastic and shear force-resistant silicon film as an interface to solve the separation force problem. However, the separation force reduction was insignificant, especially at the low-thickness interface. The slope of the separation force in relation to the platform’s displacement is varied with changing the geometry, while the maximum separation force value is almost same. It has also been observed that the separation force decreases with increasing the thickness of the silicon film. Due to the obstacle of light transparency, the optimum film thickness has been selected as a maximum limit of 2 mm [40]. An oxygen inhibition layer has been formed near the PDMS surface because of the termination of the photopolymerization process in the presence of oxygen. This phenomenon helps to create a zone where the photopolymer resin remains uncured, known as a ‘dead zone’ [56,57,58]. The process methodology of dead layer formation in free radical polymerization has been shown in Figure 7.

Zhou et al. [44] developed a PDMS-based two-channel resin vat system. Photopolymerization has been performed in the channel of the PDMS film at position one, then the vat is translated to the x-direction and moves to position two 2, which is no PDMS zone (Figure 8a). Therefore, the shear force acts on the cured part to separate it from the PDMS interface instead of the tensile force in the pulling-up process. The thickness of the dead zone is 2.5 μm which is not enough to reduce the separation force to a great extent. The island window (IW) design concept has been proposed to enhance the oxygen permeability of PDMS. The acrylic substrate has been machined to form a layer structure that allows oxygen and helps to form a liquid interface on the top of the PDMS. A maximum of 370 μm oxygen inhibition layer has been measured at particular dimensions (A = 100 mm, B = 34.17 mm, C = 5 mm) of the different regions (Figure 8b). Due to the non-uniform distribution of the oxygen, the thickness of the inhibition is not same throughout the island region. The lowest thickness has been found at the center of the vat for all possible dimensions, which is one of the main limitations. The adequate printing time has been increased by 73% compared to the conventional one. Still, the surface roughness is high at the maximum printing speed of 90 mm/h [59]. Wu et al. [60] developed a slippery ultra-low adhesive energy PDMS (S-PDMS) surface by inspiring the natural phenomena of the slippery water layer of the pitcher plant. The experiment shows an almost 13-fold reduction in the separation force compared to the standard PDMS surface. The rigid material quartz has been taken to see the effect on the separation force and the printing process. The separation force increases almost ten fold compared to PDMS, and also challenging to separate the cured layer. The thickness of (100–200) μm perfluorocarbon in S-PDMS creates nanoscale holes and worm-like channels in microsize, making the solid–solid separation into solid–liquid–solid separation. The service life of the S-PDMS is the best of these. However, the surface roughness of the cured part’s sidewall cannot be controlled. The commercial LCD VPP manufactured company Nexa 3D invented a frictionless silicon-based polymer to develop a self-lubricating interface [61]. The particle of the silicon oil has made a thin lubricating inhibition layer which eliminates the direct contact between the cured part and the interface. It has been shown that each layer’s printing speed increases almost two fold compared to the standard process with good printing accuracy. However, the area of the printing part and service life of the interface are not mentioned, which are important factors regarding mass production. Another interface modification has been patented to make an oxygen inhibition layer on the top of the interface by using 5% vitamin E with PDMS [62]. Anti-oxidant nature of vitamin E helps to release oxygen which reacts with the free radicals of the photopolymer during photopolymerization and helps to develop the dead zone of the thickness of 30 μm which helps to maintain the uncured resin on the top of the interface. However, the durability of the interface is not enough for all kinds of photopolymer resins, especially which have a high percentage of photoinitiators and monomers like isobornyl acrylate (IBOA).

Instead of using PDMS, an oxygen inhibition layer or ‘dead zone’ thickness of 30 μm has been developed by supplying pure oxygen beneath the amorphous fluoropolymer (Teflon AF 2400) window (Figure 9a), known as continuous liquid interface production (CLIP) [63,64]. Dead zone thickness depends on the purity of the oxygen, photoinitiator concentration and its absorptivity, the number of incident photons at the image plane per area at a unit of time, and resin reactivity. In CLIP, the printing speed has been controlled by the photopolymer resin viscosity, curing time, and dead zone thickness rather than the 3D modeling slicing thickness. This phenomenon helps to reduce the anisotropic nature of the printed part, which is a challenge for layer adding process. The printing speed has reached up to 1000 mm/h and 300 mm/h at 300 μm, and 100 μm layer thickness, respectively. However, the printing of macro-sized solid sections at a high printing speed is still challenging due to the difficulty of the resin flow at the printing zone. The constant flow rate of pure oxygen is necessary to maintain the constant dead zone thickness, which is another challenge, especially for large-area printing. Wang et al. [65] developed a computational fluid dynamic (CFD) model to investigate the resin flow behavior in the dead zone during printing. It has been observed that the magnitude of vacuum pressure varies with the printing area, dead zone thickness, lifting speed, and acceleration. Of these the parameters, the vacuum pressure is proportional to the fourth power of the printed part’s radius. In addition, it has been found that the vacuum pressure is maximum at the center of the part geometry in all cases. Walker et al. [66] developed the high-area rapid printing (HARP) technique to reduce the separation force and control the heat generation during the vat photopolymerization. An omniphobic fluorinated oil has a higher density than a photopolymer resin. It has been used on the glass window to make a slip boundary which allows the continuous retraction of the cured part (Figure 9b). The mobility of the oil helps to reduce the surface temperature of the printing part at the cured zone by 40% compared to static oil. The reduction in the temperature helps to control the part’s dimension stability, unwanted deformation, and other demerits. The printed parts exhibit an isotropic nature at the different orientations of the printing and have better strength than the injection molding part. Despite having so many improvements, the surface roughness of the printed part is almost 35 µm at the part’s size of 3 mm. Additionally, the higher viscosity and high contractility resins are not suitable for this process.

The semi-flexible film and flexible film states have been developed by using single and double flexible fluorinated ethylene propylene (FEP) films, respectively, to increase the resin flow into the vacuum [67]. In general LCD vat photopolymerization, the FEP film (interface) is placed on the top of the LCD screen, partial adhesion occurs due to the presence of moisture and interaction. To reduce this adhesion, another film has been inserted between them to get the flexible film state (Figure 10a). In the semi-flexible film state, an increment of 48.4% and 59.5% have been observed compared to the flexible film state for the separation force and the separation energy respectively. The bilinear CZM model with mode I and II fracture has been considered for simulation model development. It has been observed from the simulation result that the fracture initiates from the periphery of the cured part and gradually spreads to the center until the complete separation for both models. The flexible state helps the resin flow easier which reduces the viscous flow effect of the resin, thus reducing the separation force occurs. Though the method is simple and economical, the improvement in the printing speed is not significant. Furthermore, an extraordinary soft hydrogel [51,68] tuned via the crosslinker has been used as an interface for rapid printing of larger cross-sections (Figure 10b). Due to the low elastic modulus of the hydrogel, the JKR-like layer separation occurs which follows the gradual crack propagation, shown in Figure 10c. This phenomenon helps to reduce the suction force caused by the refiling of the photopolymer resin, especially for sizeable solid body printing. The maximum separation force can be reduced up to one-third of the commercially used FEP at the same printing speed. The chemical stability of the hydrogel is better than PDMS except for the pure polyethylene glycol diacrylate (PEGDMA) monomer. The minimum interface film thickness should be kept at 4 mm to obtain a lower separation force, which is not standard for VPP. The effect on the printing process because of the water affinity of hydrogel needs to be studied in detail.

### 3.3. Process Modifications

The above discussion was based on the interface/film modification studies. The experiments also have been conducted to modify the DLP and LCD VPP process for reducing the separation force by additional mechanisms like tilting the photopolymer vat, applying vibration at moderate frequency, and increasing the vat temperature to reduce the viscosity of photopolymer resin.

The tilting mechanism has been used by industries like ETEC Inc. [30] and Miicraft [35] to reduce the separation force. The motion has been applied on the platform (vat) by lifting or lowering one side of the platform and pivoting the other side. Wu et al. [69] derived an analytical model based on bilinear CZM from understanding the effect of tilting motion on cohesive stiffness and separation energy. An experimental setup has been developed (Figure 11a) to monitor the influence of the exposure area and inert film types (FEP, PDMS) on cohesive stiffness and separation energy. The tilting mechanism facilitates the peeling process by which shear force acts to separate the layer instead of the tensile force. The separation force of the tilting mechanism has been reduced by 20 percent compared to the pulling-up process by using PDMS at the separation velocity of 0.2 mm/s. It has been observed that the stiffness and separation energy increase in different magnitudes for both film types with increasing separation velocity. However, the tilting angle is dependent on the printing area, which makes it difficult for large-area printing, and extra processing time also needs for tilting. The vibration-assisted system has been developed by breaking the vacuum state between the cured part and the interface [70]. The loudspeaker of 60 Hz frequency with 50% of the maximum volume has been used to make a low-frequency vibration of the resin vat’s glass window. The hardware of the setup and schematic view of the mechanism has been shown in Figure 11b,c. The separation velocity depends on so many factors like vibration signal frequency and amplitude, system stiffness and mass, material property, and geometry of the contact area. The separation force has been reduced by 60% by using the vibration-assisted system irrespective of the printing area, which is one of the significant advantages. However, the stability of the small feature size, like 300 μm or less, is not satisfactory compared to the standard process. The repeatability of the cyclic load should be further increased. Xu et al. [71] used piezoelectric actuators to generate a more accurate cyclic load at high frequency to separate the cured layer from the interface. A stress-based fatigue model has also been chosen to develop the prototype (Figure 11d) and analytical model. Several factors like frequency, amplitude, pre-stress, exposure area sizes, geometric topologies, and geometric shapes have been reported and the relationship between the separation time and these factors has been studied. It has been observed that the separation time decreases up to 500 Hz. After that, the separation time increases due to the reduction in the cyclic force amplitude. The pre-stress also reduces the separation time by reducing the cycle of separation’s number except at higher stress amplitude. Sharp edges of part geometry also play an essential role in reducing the separation time. So, the pentagram has the minimum separation time among the circle, ring, square, triangle, and pentagon. The separation force has been reduced by around 75% compared to the direct pulling-up method. The printing of the same hourglass has been performed by using tilting separation, direct pulling-up, and vibration-assisted separation methods. Of the others, the vibration-assisted separation method is carried out successfully; however, it takes a longer separation time. 

The separation force is linearly dependent on the viscosity of the photopolymer resin. Therefore, the highly viscous resin-like ceramic slurry or adding ceramic particles to the polymer is challenging to print. The hot lithography VPP process has been carried out with two different viscous polymer resins at different temperatures (25 °C, 35 °C, 45 °C and 55 °C) [72,73]. With the increment of the temperature, the viscosity of both resins decreases, and shear thinning phenomena start to occur from 35 °C. However, the separation force is lower in the case of highly viscous resin at the higher printing temperature (45–55) °C. This result implies that the crosslinking degree and reactivity of the monomer are more dominating than the viscosity of the photopolymer resin at higher temperatures.

Based on different types of interface and process modification approaches, Table 3 and Table 4 are summarized the notable improvements and limitations in the proposed methods to reduce the separation force.

## 4. Photopolymer Resin Refilling

During photopolymerization, the liquid resin has been cured at a particular layer thickness, and fresh resin from the uncured region should be filled up in the vacuum for the next layer of printing. Incomplete filling causes high vacuum pressure which causes high separation force, cracks, and voids between layers [74,75,76]. Therefore, the high refilling time has been preferred to get a reliable product [77]. However, it increases the total printing time which should be minimized to achieve high-speed VPP. The refilling time depends on the viscosity of the resin, the part’s size, the pressure gradient, the gap between the cured part and the interface, and the part geometry [38,78]. From the study of high-speed 3D bioprinting [79], the motion of fluorescence microbeads of the hydrogel solution has been tracked to understand the flow velocity of the hydrogel beneath the cured part (Figure 12a,b). Due to the variable nature of the resin properties and part geometry, most of the studies have been conducted to develop interface properties and process modifications that can increase the resin refilling speed.

He et al. [80] presented an interface modification approach has been presented by texturing the PDMS-coated interface layer. Radial microgrooves of different sizes on the PDMS surface have been achieved by laser micromachining, shown in Figure 13a. Different sizes (20 mm to 50 mm) of models are printed on the textured surface and compared the separation force with the models which are printed on a standard smooth surface. The separation force for every design of microgrooves has been reduced by 60% compared with the smooth PDMS interface. However, the photopolymer resin in the microgrooves is cured during photopolymerization and printed on the surface ridge shape pattern, causing the surface roughness of the printed part. The average surface roughness has been found to be around 25 μm in the case of 68 mm diameter and 10 mm thick gear part, which is one of the limitations of this process. A theoretical model has also been developed based on the groove geometry, depth, and number of grooves to observe the effect on the refilling time and separation force [78]. It has been observed from the theoretical and simulation results that the separation force exponentially decreases with the increasing refilling time, groove depth, and the number of grooves. From the geometry viewpoint, the rectangular groove is more effective than the triangular shape due to the less volume of the triangular ones. However, the significant value of surface roughness and lower printing accuracy due to the interference of the transmitted light are critical issues. Wang et al. [81] developed a bio-inspired nano-sized texture on the PDMS interface bonded with the permeable membrane (Figure 13b). The hydrophobic nature has been improved because of nanotexture which helps to reduce the printing time by approximately 25% with a large cross-sectional area. Resin flow velocity has been increased by two fold compared with the flat surface and also reduced the chances of vacuum formation. The simulation model has been developed based on a two-phase laminar flow which also satisfies the experimental result. The durability of the nanostructured after repetitive printing has not been studied.

Resin refilling is also one of the critical issues in continuous printing, especially for solid cross-section printing. The standard DLP VPP process has been modified by additional horizontal translation of the vat to accelerate the resin reflow velocity during the mask video projection-based stereolithography (MVP-SL) [82]. The resin vat is mounted on a horizontal linear stage, sliding continuously along with the vertical movement (V) of the platform. However, there is no relative motion between the vat and the DLP projector. Based on the Hele-Shaw flow, a simplified exponential function of the flow speed ($V_{r}$) of unit liquid in the refilling zone has been described by Equation (6) for the square cured part of side length L. The valid side length ($L_{r}$) of the cured part can be calculated based on Equation (7) where resin viscosity (μ), printing speed (V), pressure difference (∆P), and dead zone thickness (h) are the controlling parameters. During photopolymerization, the increment of resin viscosity can be expressed by the constant q, which is zero at no absorption of light energy, otherwise non-zero constant. The printing time has been reduced by five fold compared to the image-based projection without stair-stepping defect. It has been mentioned that the limitation of the CLIP process for large cross-section areas can be solved by using this two-way movement. Further extending, Lichade et al. [83], proposed the gradient light video projection-based stereolithography (GLVP-SL) to reduce the refilling issue by dividing each binary image into three mask images with different grayscale distributions. The resin flow, exposure energy, and curing depth are measured at different grayscales. The effect of the grayscale values has been studied. The sequence of the grayscale images has also been optimized to obtain a void-free solid structure. The porosity and surface roughness have been decreased by 75% and 80% compared to the MVP-SL. However, the effect of different resin materials and part sizes larger than 10 mm has not been studied.
(6)$$V_{r}=\frac{∆P}{\mu e^{q}L}\left[{\left(\frac{h}{2}\right)}^{2}-{\left(z-\frac{h}{2}\right)}^{2}\right]$$
where, z = 0 at the interface and z = h at the cured bottom surface (shown in Figure 6a).
(7)$$L_{r}={\left(\frac{2∆{Ph}^{3}}{3\mu e^{q}V}\right)}^{\frac{1}{2}}$$

## 5. Challenges with High-Speed VPP Image Projection Systems

The LCD with LEDs as a backlight module and DMD in the digital light projector are the critical components of the dynamic mask image projection system of LCD and DLP VPP, respectively. In a system with a high printing speed and large area mask projection, the print model’s dimension tolerance and surface finish of LCD VPP are highly influenced by the LCD pixel size, light transmission efficiency, aperture ratio, polarization film, and uniformity of LED backlight. On the other hand, DLP VPP has the limitation of the power transmission and fixed pixel ratio of a DMD, especially for large-area printing.

### 5.1. LCD VPP Image Projection System

#### 5.1.1. LCD Panel

Of all other LCD technologies, a thin film transistor (TFT)-based LCD is the most popular one and use in the LCD VPP due to the high image sharpness, fast refreshing, and low energy consumption. The TFT-based LCD (Figure 14a) [84] can be divided into amorphous silicon (a-Si), continuous grain silicon (CGS), low-temperature polycrystalline silicon (LTPS TFT), and indium-gallium-zinc-oxide (IGZO) technology [85,86]. The light transmission efficiency through LCD linearly depends on the aperture ratio which can be defined by the ratio of the light transmission area of the pixel to the total area of the pixel. Most of the commercially available LCDs have red, green, and blue (RGB) sub-pixels in one pixel. However, red and green pixels are inactive for the LCD VPP commonly because of using a 405 nm UV LED backlight. Therefore, the aperture ratio has been decreased by 3 fold, which reduces the light transmission efficiency, lifetime of the LCD, and prolonged curing time. Monochrome LCD has been used to solve this problem by removing the color filter which increases the output light transmission efficiency from 1% to 6% which helps to reduce the overall printing time. However, the availability of monochrome LCD panel sizes larger than 15 inches is difficult because most of the manufacturers still focus to develop larger RGB LCD panels for televisions, monitors, etc., which is not suitable for VPP [87].

Polarizer films act like a filter that allows light in a particular pattern or orientation. These films (polarizer x and y) are oriented orthogonally to each other and placed LCD screens in between them (Figure 14b). Due to the surface reflections and parasitic absorption, the light transmits only 25% and 44% of the total incoming light at 405 nm wavelength in the case of the film and wire grid polarizer, respectively [88,89]. In most cases, stretched sheet polyvinyl alcohol (PVA) has been used for film polarizers which absorb the reflected light, resulting in overheating at lower light intensity. Due to this problem, a wire grid polarizer has been used to increase the monochrome LCD output intensity by more than 50 mW/cm^2^ [90]. This system helps to cure the low-sensitive material like PCHS-silicones. However, the manufacturing cost of such wire grid polarizers is reasonably high at lower wavelengths like 405 nm. An array of LEDs has been used (Figure 14c) as the LCD backlight to achieve uniform exposure over a large area. The emission of the LED array should be collimated to achieve defect-free printed parts irrespective of the LED’s wavelengths [91,92].

**Figure 14 polymers-15-02716-f014:**
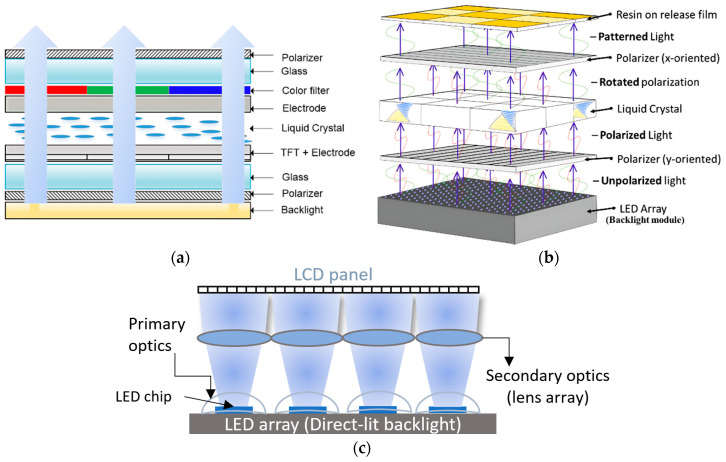
(**a**) Schematic representation of the standard RGB LCD. Reproduced from [84]. (**b**) Diagram of the unpolarized light of LED array passes through components in LCD VPP. Reproduced with permission from [90]. (**c**) Schematic arrangement of the LED’s primary and secondary optics in standard LCD VPP.

#### 5.1.2. Primary and Secondary Optics of LEDs

The packaging of LEDs or primary optics and the secondary optics (Figure 14c) have an important role in increasing LCD panels’ optical efficiency and irradiance homogeneity. The monochromatic, Gaussian-shaped emission spectra of LEDs have a direct relationship with their solid-state semiconductor light generation mechanism. The peak wavelength of the LEDs depends on the semiconductor chip’s bandgap energy, which is equal to the emitted photon energy (E). The wavelength (λ) can be determined by Planck’s equation (E = h’c/λ), where h’, c are Planck’s constant and light speed, respectively [93]. The gallium nitride (GaN)-based LEDs revolutionize UV-LED technology to produce a wide range of UV irradiation by changing their composition [94]. A summary of the effect of the primary lens doping material and substrate modification for near UV (NUV) LEDs is presented in Table 5. The low power consumption, fast response time, compact size, low cost, and high stability LEDs are very much essential as LCD backlight modules to achieve the high-speed LCD VPP [95,96,97,98].

Silicone lens has been considered as primary optics in LED packaging which controls the emitting light efficiency and uniformity. A bio-inspired primary lens has been developed with periodic external nanostructures (Figure 15a). The maximum transmittance has been increased up to 98.3%, and the optical efficiency can be enhanced by 3%, in comparison with the conventional smooth lens surface [99]. Su et al. [100] proposed a novel packaging structure to increase the optical efficiency of LED packaging. The scattering material

TiO_2_ nanoparticles are doped into the silicone lens, and developed five different dual-layer package structures to compare the result. Of these package structures, a thinner TiO_2_-doped silicone layer around the LED chip and a pure silicon lens on top (Figure 15b) are the most effective ones. The LED module’s optical efficiency has been increased by 24.5% compared to the conventional single-layer encapsulated LED chip. A crater-feature silicone lens along with quantum dot (QD) film has been used in packaging (Figure 15c) to increase the optical efficiency by 70% at a high driving current and 50% at a low driving current compared with the traditional hemispherical lens [101].

**Table 5 polymers-15-02716-t005:** Summary of the effect of doping materials and substrate modification of NUV LEDs’ primary optics.

Methods	Noticeable Improvement	Ref.
Eutectic flip chip NUV LED with a aluminum nitride nanoparticle (AlN NP)-doped silicon layer as primary optics material	The Light output power has been increased by 17.4% with a reduction in junction temperature by 5.7 °C compared to the silicon layer.	[102]
Controlling the eutectic voidage of eutectic flip chip NUV LED (void reduction from 30% to 3%)	The Light power has been enhanced by 13.3% with a reduction in junction temperature by 3.5 °C due to the void reduction from 30% to 3%.	[103]
Chip on board (COB) packaging module with microlens array (MLA) of silicon has been used as the primary optics design	The light extraction efficiency (LEE) has been increased by 8% compared to the without MLA.	[104]
Spin on glass (SOG) MLA primary optics in circular shapes of different diameters	The LEE has been increased by 21.86% when using the 50 µm diameter.	[105]
Embedding The air gap between the LED chip and substrate by 4 wt% graphene oxide (GO)-based silicon composite	The junction temperature and internal thermal resistance have been reduced by 2.9 °C, and 34% respectively.	[106]
NUV LED Sapphire substrate modification by ultra-thin air cavity nanopatterned	The LEE has been increased by two fold compared to the flat sapphire substrate.	[107]

The secondary optics are equally important to collimate the high divergence output light from LED primary optics. The free-form optics in terms of design freedom of the lens curvature provides better control of light irradiation and high-quality LEDs [108,109,110,111,112,113]. Additionally, the direct output from the LED package is normally a circular spot which causes large energy loss due to the rectangular pixel of the LCD panel [114,115]. Kumar et al. [116] developed a novel free-from silicon optical lens of 23 mm diameter and 19.03 mm thickness by using the three freeform refractive surfaces (AB, GE, GH), spherical refractive surface (BC), and the ellipsoidal reflective surface (CD) where all surfaces are axially symmetric to the optical axis (Figure 16a). A Lambertian patterned UV-LED with 365 nm wavelength, 120 deg. divergent angle, 0.96 W power consumption, and 1.25 mm square chip have been considered as a power source to do the simulation. The lens has been placed 200 mm away from the source, and the angle distribution is calculated at ±2° by full-width half-maximum (FWHM) with an optical efficiency of 58.82%. This divergence angle can be reduced more if the source is considered a point. A recent study has been carried out by designing the collimated lens by using commercially available PMMA material, especially for UV at 405 nm wavelength [117]. The commercially available six different UV LEDs with six different divergent angles based on FWHM are chosen to conduct the simulation. The lens for individual LED has been designed based on the distance between the actual LED light source and S1($r_{0}$), the distance between the virtual light source and the LED (${\xi r}_{0}$), and the central thickness, where S1 and S2 are the two aspherical lens surfaces (Figure 16b). The analytical results have shown that the high divergent irradiation transfer to 1.56° to 2.84° based on FWHM along with satisfactory optical efficiency. However, the experimental measurement has not been conducted to validate the numerical outputs.

There are several other methods applied to collimate the wide-angle LED illumination by designing the microfeatures with an inverted prism film [118,119], two groups of microprism array [120], a waveguide with pixelated nanogratings [121] of the edge-type backlight. However, the direct-lit LED backlight module has been used for LCD VPP due to the advantage of local dimming and size scalability to increase the contrast ratio [122]. Lin et al. [123] experimented on a 13-inch LCD VPP projection system to enhance the irradiance homogeneity of LCD panels by using the diffuser, brightness enhancement films (BEF), and 1D privacy filter for direct-type backlight (Figure 17a,b). The LCD panel uniformity, surface roughness, and dimensional accuracy of the printed part have been compared with the commercially available lens array module. The ten-point intensity measurement process has been conducted on 8.9 inches LCD panel. The uniformity of the direct-lit LED array has improved from 56% to 80% compared with the commercial lens array module. It has been observed that the intersection area of the individual lens array module is the main reason for this non-uniformity. Despite the improvement of the uniformity along with the surface finish of the printed parts, the light output intensity from the LCD panel reduces up to 5 fold compared with the lens array model, which is one of the limitations for further improvement.

### 5.2. DLP VPP Image Projection System

A digital micromirror device (DMD) is a binary pulse-width light modulator, a core element of DLP VPP. It consists of millions of switchable microscopic mirrors arranged in a rectangular array equivalent to the projected image’s pixels. A schematic view of a single pixel of DMD with important components situated on a complementary metal-oxide semiconductor (CMOS) is shown in Figure 18a. During the illumination of the light beam, each micromirror can rotate with the help of mechanical actuators either +12° or −12° to the surface normal to define the on-off states (Figure 18b) in the standard process [124,125,126]. The largest available size of high-resolution DMDs is 0.9 inch at 4K resolution having 7.6 µm pixel size, maximum power of 26.6 W at 400 nm to 420 nm light wavelength [90,127]. The mentioned power is reduced to 6W at the wavelength ranging from 363 nm to 400 nm. If it uses to project a 30-inch image, the pixel size increases to 250 µm due to the fixed ratio of pixel size to image projection ratio [128]. However, a pixel size of 158 μm is available in a 10-megapixel 30-inch LCD monochrome panel at a lower cost. Apart from this, the intensity of the DMDs would be 1.4 mW/cm^2^ for the above situation which increases the standard resin photopolymerization time. Moreover, the manufacturing of the DMD is the core of the trademarked DLP projection technology from Texas instruments company [129]. Despite that, there are several methods have been invented to overcome this technical constraint by which DLP VPP can be used for large-area printing to maintain high image resolution.

Emami et al. [130] developed the scanning projection stereolithography (SPSL) method by adding the scanning and projection methods by which both methods are carried out simultaneously. This is different from the ‘step-and-repeat’ [111,131] method where the scanning and projection have been performed one after the other, increasing the overall printing time and polymerization overlapping chance. A schematic and experimental setup of the SPSL method has been shown in Figure 19a. This process has a similarity with video animation frames so the scanning speed is an important control parameter. If the scanning speed is too fast the resin cannot be cured properly on the other side too slow scanning speed causes over-curing. The reduction in the surface roughness and adequate lateral strength is challenging for the SPSL due to the movement of the optical projection module. To resolve the projector movements issue, the multi-projector DLP with energy homogenization has been used to enlarge the build area instead of translating the projected images [132]. Output energy from the two projectors has been homogenized by using the tiling pattern of the projectors. The main concerned area is the intersection of the projected images from two individual projectors. Though the gray level values are controlled from 25 to 255 at intervals 10, it is difficult to get the black or zero value pixel at the intersection area. Wang et al. [133] developed a double mask projection stereolithography (DMPSL) system to add the large envelope projection (LEP) and high accuracy projection (HAP) modules (Figure 19b). The approach behind this setup is to increase the build size with help of the larger pixel of LEP while HAP makes the highly accurate part if necessary. The straight line and sawtooth splicing strategies have been adopted to increase the mechanical strength at the splicing interface. It has been observed that the increment of the overlapping width in terms of pixels, increases the mechanical strength. However, the bending strength has been decreased by increasing the overlapping width during the sawtooth splicing strategy while it remains the same in the straight-line method.

Without using multiple DLP projectors or an additional exposure system, the delta mechanism has been implemented to increase the build size using a single projector [134,135]. Conventionally the build platform can only translate along the vertical direction while it can move and rotate in the horizontal plane in the delta mechanism. It is also useful to obtain high or low-accuracy print by adjusting the focal length that overcomes the fixed pixel ratio constraint of DMD. By using the developed algorithm of medial-axis-driven segmentation and covering, the big-sized projection image has been cut into smaller sizes so that each smaller-sized image can fit into the projected area. Huang et al. [136] used the tripteron parallel mechanism with orthogonal 3- degrees of freedom (DoF) along three axes individually to move the projection module in a desirable position according to the dimension of the printed part (Figure 19c). The degree of movement of the projector has been increased to nine which gives a great extent of freedom to project an image at different accuracy. However, the image stitching method needs to be applied after a certain image size which can cause poor mechanical properties. The stability of the mechanical moving systems is another crucial factor to achieve high dimension accuracy in comparison with the standard DLP process.

**Figure 19 polymers-15-02716-f019:**
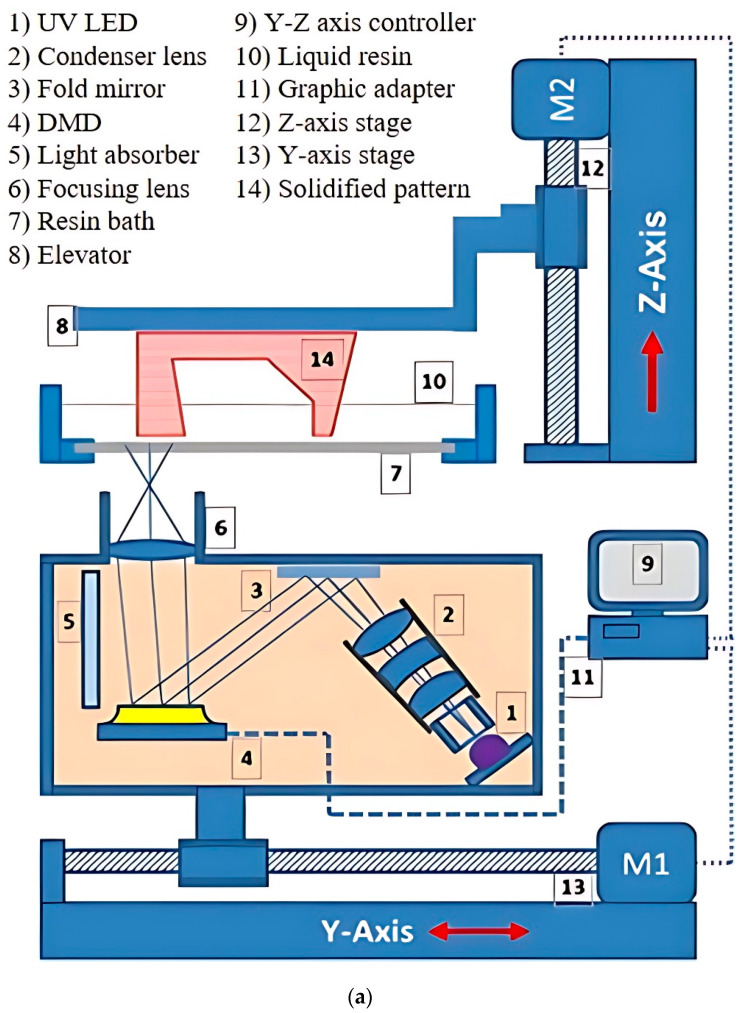
Printing area increment methods of DLP VPP. (**a**) Schematic machine setup of the scanning projection stereolithography (SPSL). Reproduced with permission from [130]. (**b**) Schematic representation of double mask projection stereolithography (DMPSL). Reproduced with permission from [133]. (**c**) Increment of the projection area by tripteron parallel mechanism. Reproduced from [136].

## 6. Concluding Remarks and Future Works

The review has been carried out to identify the critical issues of high-speed LCD and DLP VPP and summarized the appropriate studies which can provide the solutions and direction for further research to bring out the LCD and DLP VPP as successful high-speed AM processes. The term high-speed VPP has been addressed as a high volumetric print rate (Table 1) where both the printing area and speed are essential from a production viewpoint. To produce high-resolution and large area parts is still challenging because of the high separation force, the extended resin refilling time, low transmission efficiency of LCD, the problem of collimation of LEDs as LCD backlight, and limitation of light intensity capacity and fixed pixel ratio of DMDs. The following concluding remarks have been drawn based on the literature survey:A proper understanding of the separation mechanism is essential to estimate the separation force accurately whether it is solid–solid (DMT or JKR) or solid–liquid separation. For the solid–solid separation, the cohesive zone model (CZM) followed by the bi-linear traction separation law (TSL) has been used in most of the studies where crack formation occurs due to tension (mode I). It has been observed that soft material like hydrogel exhibits JKR-like separation while the higher stiffness of the interface is responsible for DMT-like separation. On the other hand, for solid–liquid separation due to dead zone, the analytical model of separation force has been developed based on the Navier–Stokes equation. The separation force is proportional to the fourth power of the radius of the printed part and inversely proportional to the third power of the dead zone thickness during solid–liquid separation.The interface modifications are the most focused area to reduce the separation force by creating an oxygen inhibition layer or dead zone. Polydimethylsiloxane (PDMS) has been used in most of the studies due to its oxygen permeability, even additives like silicon oil, perfluorocarbon, and vitamin E are also added to make a thicker dead zone. The CLIP and HARP technologies have been developed to maintain a constant dead zone thickness. However, the chemical stability of the dead zone against different monomers, the concentration of photoinitiators, and chemical additives are still uncertain. Additionally, maintaining the oxygen purity, and constant flow pressure in CLIP is indispensable otherwise the dead zone thickness will reduce drastically.The tilting mechanism has been used in several commercial machines where separation takes place due to shearing force instead of tension. Therefore, the required separation force is lower than the standard pulling process. However, this mechanism is difficult to apply for large-area printing because the tilting angle depends on the area of the vat. Furthermore, the repeatability, amplitude, and frequency of the vibration-assisted system should be explored in detail to reduce the separation force and surface roughness.During the separation, a vacuum is created between the cured part and the interface. The microand nano-sized channels of different shapes and sizes have been designed to increase the flowability of the resin into the vacuum. The surface roughness of the printed part due to the microchannel is one of the major concerns. Soft materials like hydrogel can also be another cost-effective solution to increase flowability. Chemical stability and JKR-like separation help to print large section areas at a minimum separation force. However, the thicker hydrogel (>4 mm) and water absorption tendency raise doubt about compatibility with the mass production scale.The monochrome LCD panel is essential to increase the printing speed and lifetime of the panel; however, the availability of large-sized monochrome LCD panels is still difficult. For increasing the optical efficiency and controlling the divergence of the LED backlight, the lens design plays an important role in primary and secondary optics. For the commercial lens array, the irradiance inhomogeneity of the LCD panel arises due to the intersection area among the secondary lens modules.Due to the area constraint and fixed pixel ratio of DMDs, multiple projectors methods (SPSL, DMPSL) have been used to stitch the images to increase the printing area. Otherwise, a higher DoF mechanism has been added to the DLP projector or vat to increase the printing area by adjusting the focal length. However, obtaining the desired mechanical properties and surface roughness of the stitching zone is one of the main difficulties.

Future research perspectives can focus on the following areas:Further studies on the cohesive zone model (CZM) for solid–solid separation can be conducted to incorporate the trilinear, parabolic, and exponential traction laws, to improve the correlation between numerical models and experimental results.Simultaneous consideration of separation force and resin refilling issues through the development of fluid-structure interaction (FSI) numerical model can help to understand and control the factors such as resin viscosity, printed part dimensions, and geometry, interface surface properties, dead zone thickness, especially for large-area printing.Development of ultra-slippery, low surface energy interface materials needs to be continued so they can maintain their surface properties after repetitive printing in different monomers and photoinitiator concentrations. Alternatively, exploring the hydrogel-like soft materials with various crosslinkers as interface materials could be promising.Investigation of LEDs’ primary optics lens designs needs to be explored further, specifically from a high-speed LCD VPP perspective. The different freeform shapes of secondary optics with BEF can be studied in more detail to enhance irradiance homogeneity and coherence.The concept of local dimming in mini/microLEDs can be applied to control the mask projection accurately with a reduction in power consumption and pixel blending effect in LCD VPP.

These future research directions have the potential to address critical challenges and contribute to the advancement of high-speed VPP technologies, enabling their competitiveness with traditional manufacturing processes and expanding their application capabilities.

## Figures and Tables

**Figure 1 polymers-15-02716-f001:**
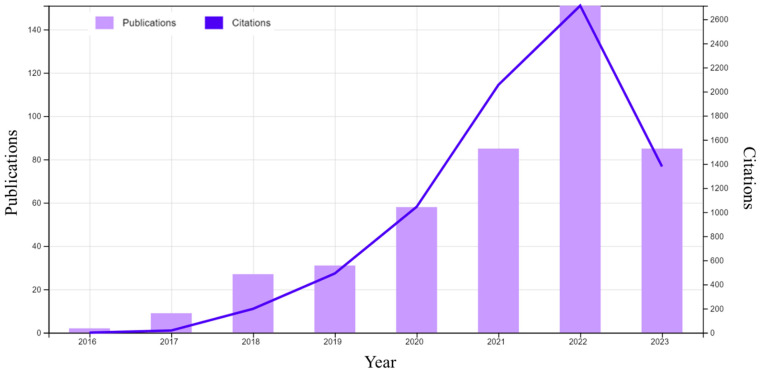
The publications and citations statistics made for VPP from the year 2016 to 12 June 2023. (source: Web of Science).

**Figure 2 polymers-15-02716-f002:**
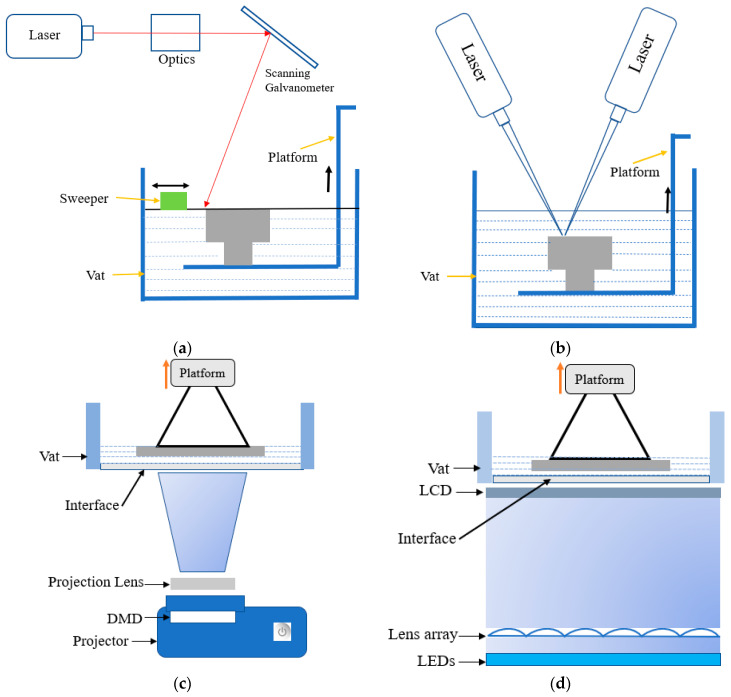
Schematic diagrams of VPP processes. (**a**) Laser-based SLA VPP. (**b**) TPP VPP. (**c**) DLP VPP. (**d**) LCD VPP. Reproduced from [13,20].

**Figure 3 polymers-15-02716-f003:**
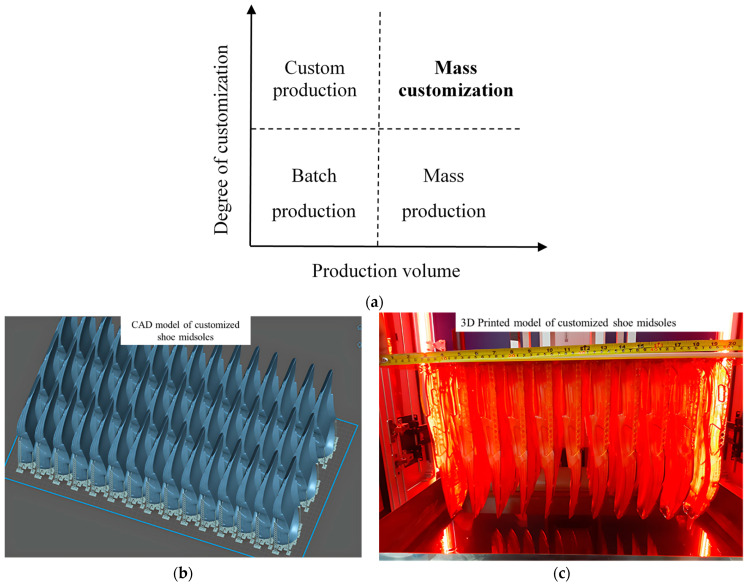
(**a**) Classification of production methods. (**b**,**c**) Mass customization by using LCD VPP to arrange near similar geometries on the same platform. (**b**) CAD models of shoe midsoles. (**c**) Printed models of shoe midsoles.

**Figure 4 polymers-15-02716-f004:**
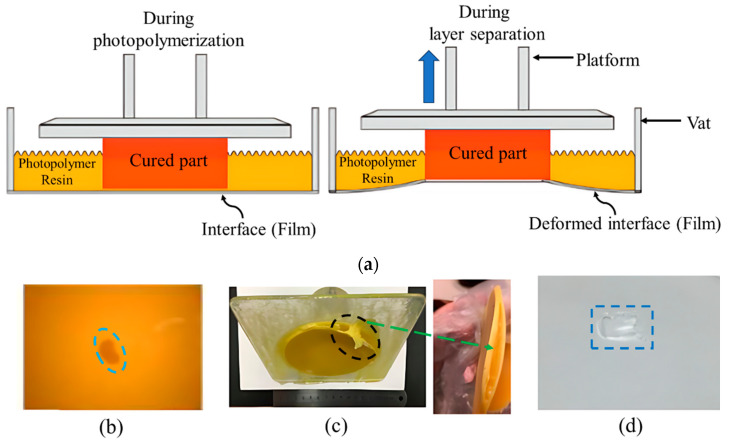
(**a**) Schematic diagram of interface deformation due to the platform’s upward movement in the bottom-up VPP. (**b**–**d**) Printing failure and defects due to the high separation force. (**b**) Holes on the printed surface, (**c**) separate printed layers, and (**d**) broken film (interface), which acts as a constrained surface. (**b**–**d**) Reproduced from [38].

**Figure 6 polymers-15-02716-f006:**
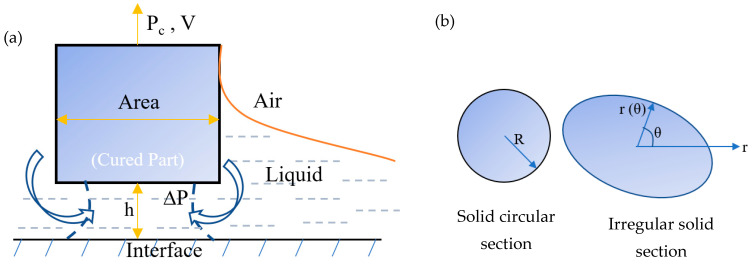
Separation mechanism for solid–liquid interaction. (**a**) The uncured resin presents in between the cured part and the interface to create the dead zone of thickness h. (**b**) Schematic view of the solid circular section and irregular solid section of the printed geometry [38].

**Figure 7 polymers-15-02716-f007:**
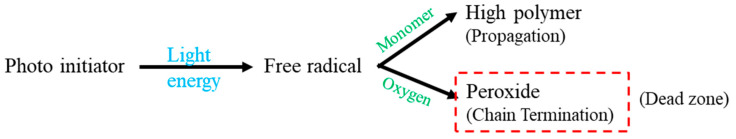
Dead zone formation in free radical polymerization.

**Figure 8 polymers-15-02716-f008:**
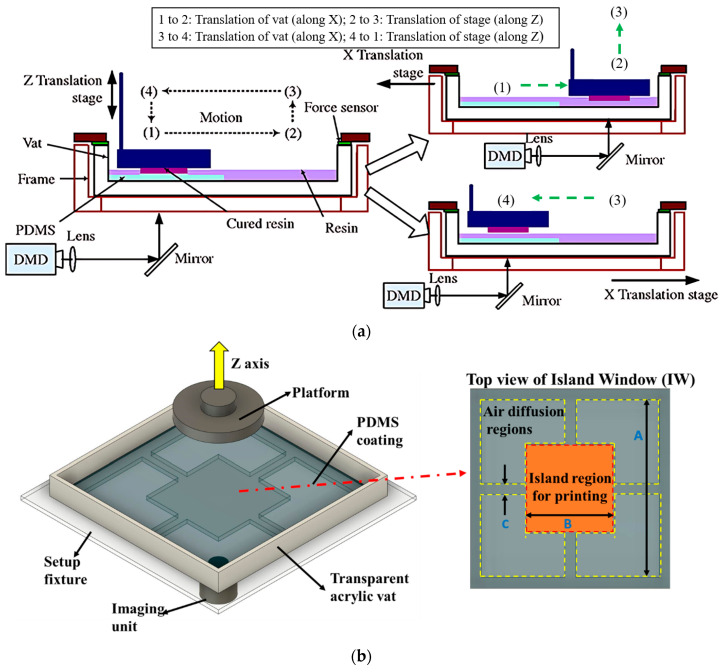
Interface modification by using PDMS in different ways. (**a**) Schematic demonstration of two-stage resin vat process. Reproduced with permission from [44]. (**b**) Schematic representation of the island window (IW) design of the vat. Reproduced from [59].

**Figure 9 polymers-15-02716-f009:**
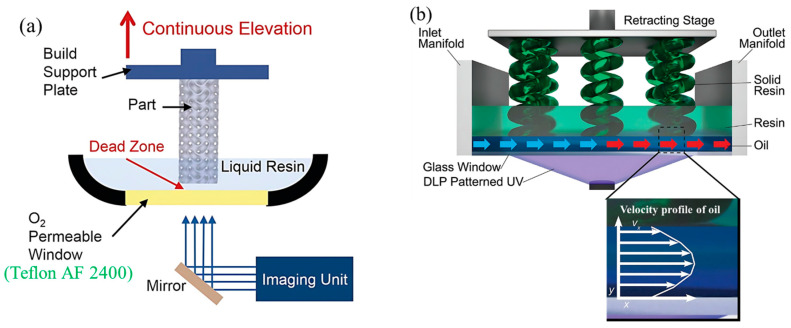
Interface modification by maintaining constant dead zone thickness. (**a**) Schematic view of continuous liquid interface production (CLIP) by supplying pure oxygen beneath the oxygen-permeable window. Reproduced with permission from [63]. (**b**) Schematic representation of the high area rapid printing (HARP) process by using the laminar flow of fluorinated oil at the top of the glass window. Reproduced with permission from [66].

**Figure 10 polymers-15-02716-f010:**
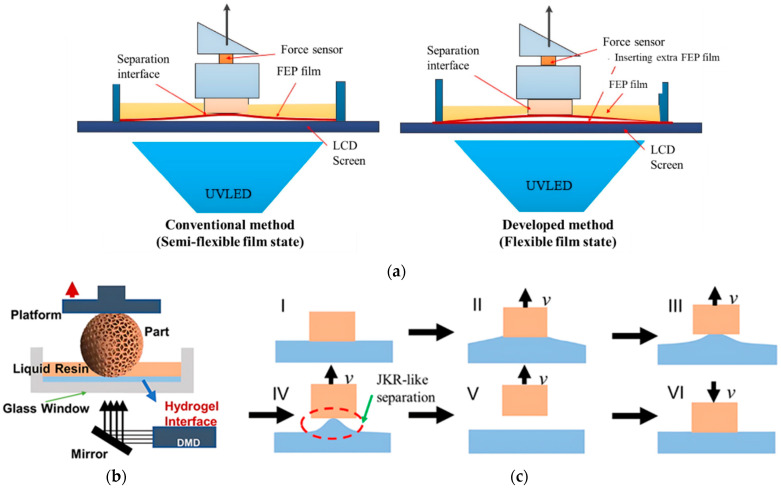
(**a**) Schematic diagram of the flexible and semi-flexible film state using FEP films. Reproduced with permission from [67]. (**b**) Schematic representation of DLP VPP setup by using the ultra-soft hydrogel as an interface material. (**c**) Schematic process of JKR-like separation on a soft hydrogel interface. Reproduced with permission from [51].

**Figure 11 polymers-15-02716-f011:**
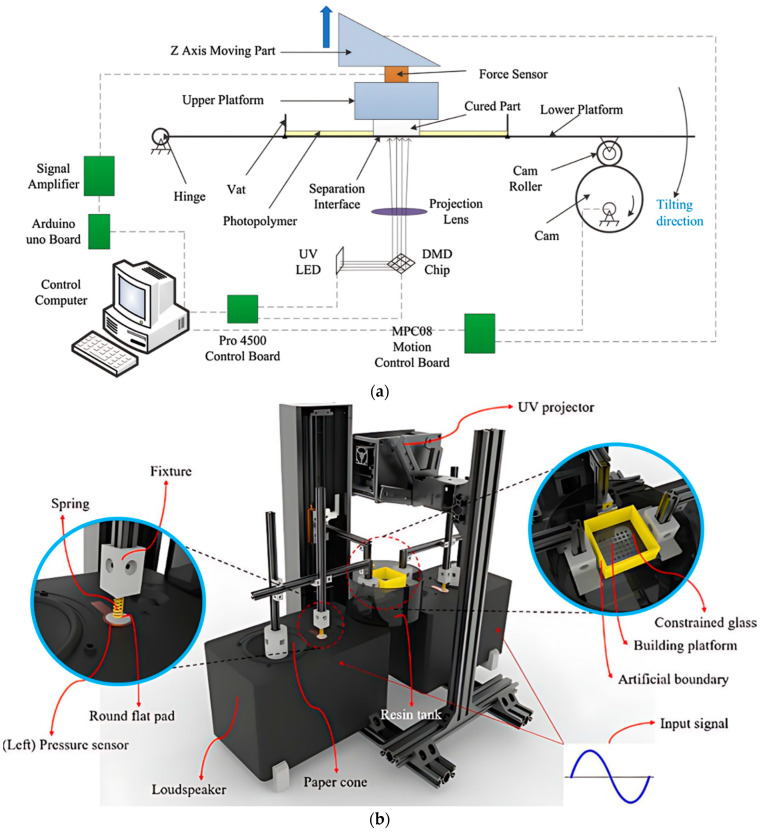
Process modifications for separation force reduction. (**a**) Schematic representation of the vat tilting motion by using a cam, roller, and hinge joint. Reproduced with permission from [69]. (**b**) Machine setup and (**c**) schematic view of the working mechanism of the vibration-assisted system using a loudspeaker of 60 Hz. Reproduced with permission from [70]. (**d**) Schematic illustration of the vibration-assisted DLP VPP machine setup using piezoelectric actuators. Reproduced with permission from [71].

**Figure 12 polymers-15-02716-f012:**
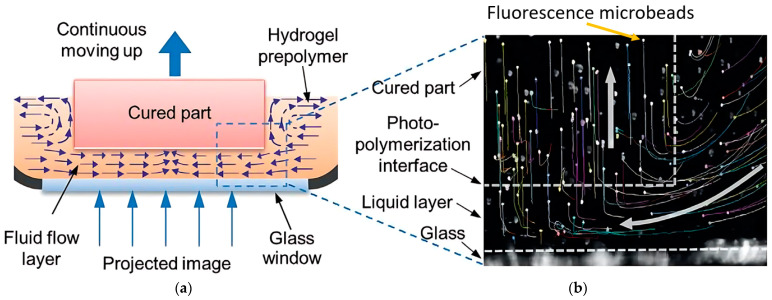
Photopolymer resin refilling for newly cured layer beneath the previous cured part. (**a**) Schematic representation of the bioprinting interface. Schematic view of the continuous replenishment of the hydrogel below the curing part. (**b**) Tracked trajectories path of fluorescence microbeads during the vacuum refilling created by previous layer printing. Reproduced with permission from [79].

**Figure 13 polymers-15-02716-f013:**
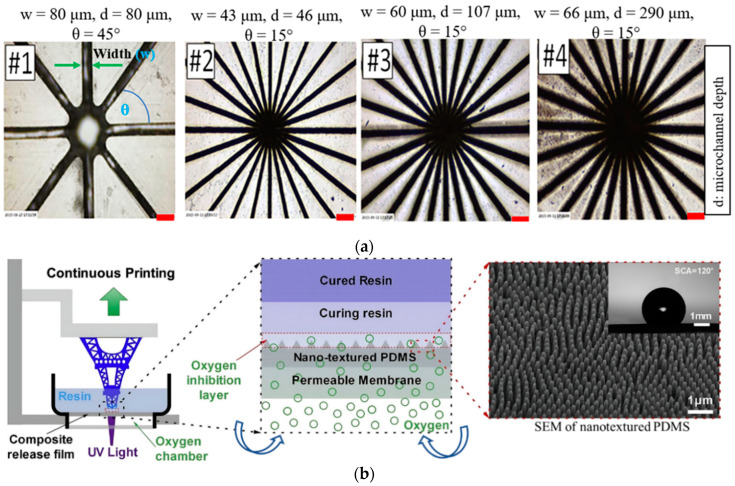
(**a**) Microscopic images of different sizes of laser-fabricated microchannels in the PDMS interface. Reproduced with permission from [80]. (**b**) Schematic and SEM view of the nanotextured PDMS interface. Reproduced with permission from [81].

**Figure 15 polymers-15-02716-f015:**
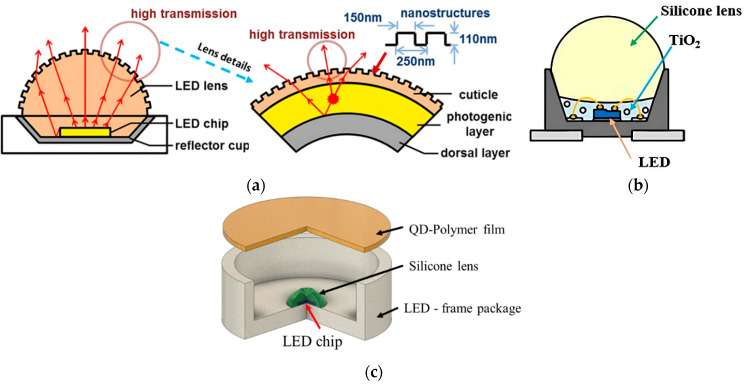
Schematic view of LED package structures (primary optics). (**a**) High-power nanostructure LED packaging inspired by firefly light organ. Reproduced with permission from [99]. (**b**) TiO2-doped silicon layer with silicone lens LED packaging. Reproduced with permission from [100]. (**c**) Quantum dot (QD)-based LED with crater silicon lens. Reproduced from [101].

**Figure 16 polymers-15-02716-f016:**
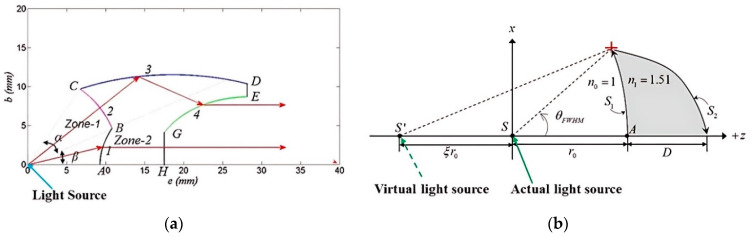
Collimating lens design of LED secondary optics. (**a**) Two-dimensional form of collimating lens consisting of three freeform surfaces, spherical and ellipsoidal surfaces. Reproduced with permission from [116]. (**b**) Two-dimensional form of collimating lens consisting of two aspherical surfaces. Reproduced with permission from [117].

**Figure 17 polymers-15-02716-f017:**
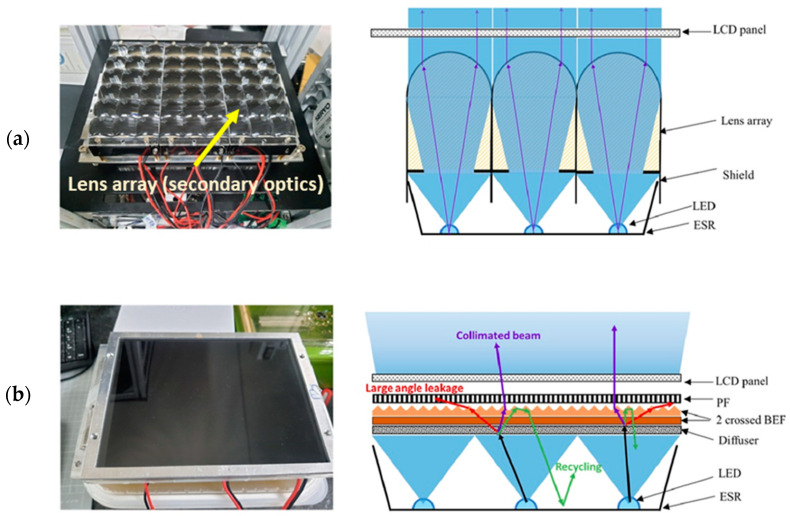
LED modules for LCD VPP. (**a**) The commercial lens array LED module. (**b**) A developed LED module by using the diffuser, BEF, and PF. Reproduced from [123].

**Figure 18 polymers-15-02716-f018:**
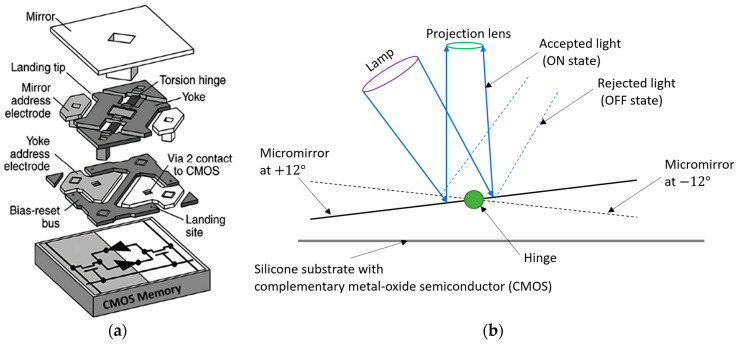
Schematic diagram of digital micromirror device (DMD). (**a**) The exploded view of a single DMD pixel consisting of a micromirror. Reproduced from [124]. (**b**) The working mechanism of the individual micromirror.

**Table 1 polymers-15-02716-t001:** Summary of the important specifications of commercially available high volumetric print rate LCD and DLP VPP machines.

VPP Methods	Projection Area (mm^2^)	Build Height (mm)	Pixel Size (μm)	Layer Resolution (μm)	Printing Speed (mm/h)	Volumetric Print Rate (Liter/h)	Company (Reference)
LCD	790 × 395	1200	91	100	16	5	Photocentric [29]
DLP	450 × 371	399	100	100	45	7.5	ETEC [30]
LCD	330 × 185	400	43	10	70	4.3	Phrozen [31]
LCD	292 ×165	400	75	50	30	1.4	Flashforge [32]
LCD	300 × 164	298	46	10	60	3	Anycubic [33]
LCD	278 × 156	300	51	10	70	3	Elegoo [34]
DLP	250 × 140	190	65	12.5	20	0.7	Miicraft [35]
LCD	195 × 115	210	52	50	160	3.6	Nexa3D [36]
DLP	189 × 118	326	75	25	300	6.7	Carbon3D [37]

**Table 2 polymers-15-02716-t002:** Summary of the effect of the printed part’s size, geometry, process parameters, and interface thickness on the separation force of LCD and DLP VPP.

VPP Methods	Interface	Interface Thickness (mm)	Cross-Sectionof Printed Part	Exposure Time (s)	Layer Thickness (mm)	Lifting Speed (mm/s)	Important Observations
DLP [38]	PDMS	2, 4	Circular, Hexagon, Triangle, Star (Constant area: 314 mm^2^)	_	_	0.63, 1.56, 2.00, 3.00	The separation force increases almost linearly with increasing the lifting speed for both thicknesses of PDMS. The measured maximum separation force decreases with increasing the PDMS thickness such that 2.44 N and 1.40 N are measured for 2 mm and 4 mm thickness, respectively, at 3 mm/s. The measured separation forces are 6.16 N, 5.87 N, 5.45 N, and 4.9 N for circular, hexagon, triangle, and star cross-sections, respectively.
DLP [39]	PDMS	2	Square (600 mm^2^)	10	0.025	0.2, 0.5, 1.0, 3.0, 5.0	The separation force reduces from 27.5 N to 2.5 N (approximately), with a reduction in the lifting speed from 5.0 mm/s to 0.2 mm/s. However, the separation time increases almost 10 fold due to the speed reduction.
LCD [40]	PDMS	0.16, 1, 2, 3	Rectangular, Circle, Triangular, Hollow circle, T-type and U-type (Constant area: 1600 mm^2^)	2	0.2	_	The maximum separation force is independent of the part’s geometry. The separation force has been reduced by increasing the interface thickness.
DLP [43]	_	_	Square (Area: 2500, 1875, 1250, 625 mm^2^)	_	0.025, 0.05, 0.1	_	The separation force increases linearly with increasing the area for every layer thickness.
DLP [44]	PDMS	1	Triangle, Square, Hexagon, Circle, T-shape, U-shape, Band, Star-shape (Constant area: 625 mm^2^)	1, 0.5, 0.15	-	0.6	The longer exposure time increases the separation force irrespective of the geometry. The separation force deviation among the geometries decreases with shorter exposure time and smaller image projected area. Of these, the U shape generates the maximum separation force (~36 N) at 1s exposure.

**Table 3 polymers-15-02716-t003:** Summary of the DLP and LCD VPP improvement by the interface modification methods.

VPP Methods	Modification Method	Noticeable Improvement	Limitations
DLP [44]	Two channels resin vat system based on PDMS coating	The layer Separation occurs due to the shearing force instead of the pulling force. The separation force of 2 channel systems is only 4–5% of the standard one, irrespective of the image shape and exposure time.	The vat size has been increased two fold, which is not an efficient solution for large-area printing. The overall process time has been increased due to additional platform movements.
DLP [59]	Island window (IW) resin vat based on PDMS coating	The dead zone thickness increases by more than 200 µm. The separation force has been reached at the minimum value at the maximum size of 34.17 mm square IW design. It can maintain the separation force below 0.5N up to 3500 s while conventional PDMS design can sustain up to 2000 s approximately. The maximum printing speed (22 cm diameter cage model) reaches up to 80 mm/h by using a 34.17 mm square IW design.	The dead zone thickness is not uniform throughout the IW. The minimum thickness has been found at the center. The oxygen permeation rate is slower than consumption which creates an unstable dead zone. The active print area is less than 50% of the total vat area.
DLP [60]	Perfluorocarbon-based PDMS interface (S PDMS)	The reduction in separation force by 13 fold compared to the PDMS, silicone oil swelled PDMS, and there was a ten-fold reduction compared to fluorinated PDMS. The separation force increment is negligible up to UV light intensity of 25mW/cm^2^ and 20 mm/min lifting speed. Resin refilling speed is two-fold faster than other surfaces.	The preparation of the S PDMS surface needs special treatment, like nitrogen gas blowing. Furthermore, it needs to be stored in the perfluorocarbon after use. The rough surface of the printed part has occurred while using polyurethane acrylate resin.
DLP [38]	PDMS coating on porous transparent plate	Separation force reduces up to 10 fold compared to the conventional method.	The reliability of the film has not been mentioned. The oxygen permeability rate must be checked after a certain period. The uniformity of light intensity needs to be checked at the vat projection surface.
DLP [63,64]	Teflon AF 2400 with externally oxygen supply (CLIP)	The dead zone thickness (20 to 30 μm) can be maintained throughout the operation, which enables continuous printing. The printing speed can be obtained up to 300 mm/h and 1000 mm/h at 100 μm and 300 μm layer thickness, respectively.	The dead zone thickness highly depends on the oxygen pressure and purity level. The resin photoinitiator concentration and absorptivity control the dead zone thickness. Large cross-section solid body printing is still challenging.
DLP [66]	Mobile liquid interface by using fluorinated oil (HARP)	The issues related to the temperature can be controlled during the continuous printing. The anisotropy has been reduced to a great extent, with even better performance than the injection molding part. The printing speed increases to 432 mm/h at 100 μm layer thickness.	The controlling of the oil flow rate is a critical issue. The printing speed reduces to 108 mm/h for butadiene rubber resin. The surface roughness increases linearly with part size. The roughness value reaches 35 µm at 3mm dimensions.
DLP [51]	Soft hydrogel tuned via the crosslinker as interface	The separation force is one-fourth compared to FEP. The printing speed can rise from 200 mm/h to 400mm/h when changing the diameter of the cylinder from 10 mm to 0.5 mm. The interface preparation is cost-effective and applicable for large-scale VPP.	Castable photopolymer resin contains PEGDMA, which is incompatible with hydrogel. Due to the high tendency of water absorbance, the durability of such an interface needs to be studied in detail.
LCD [67]	Double FEP films	The average separation force has been reduced by 42.5% compared to the single FEP. The deflection of the separation film is almost 25% less than a single FEP. The method is cost-effective, and modification can be performed quickly.	The printing speed has not increased significantly. The light intensity uniformity at the vat surface needs to measure at double FEP due to the high divergence of the LCD panel.

**Table 4 polymers-15-02716-t004:** Summary of the DLP and LCD VPP process modification methods to reduce the separation force.

VPP Methods	Modification Strategy	Noticeable Improvement	Limitations
DLP [69]	Resin vat tilting (using cam follower mechanism)	The separation force is reduced by 20% compared to the pulling-up process when using PDMS as an interface at 0.2 mm/s lifting speed.	The stress distribution in the vat along the tilting axis is not uniform. At the hinge, the stress is minimum and increases away from it. By using this method, the separation force increases by almost 11% compared to the pulling up when using FEP as an interface.
DLP [70]	Resin vat Vibration (using loudspeaker)	The measurement of maximum separation force does not go beyond 200 gm while it reaches up to 500 gm during the normal separation.	The print failure rate is higher than the normal process when the feature size ≤ 300 μm. The reliability of the vibration source needs to improve.
DLP [71]	Resin vat vibration (using piezoelectric actuators)	The reduction in the separation force is around 75% for different cross-section areas compared to the pulling-up process.	The separation time takes longer than the standard process. The printability of smaller dimensions (<10 mm) has not been checked.
LCD [72,73]	Hot lithography (increasing resin temperature)	A significant viscosity reduction in approximately 300 mPa.s can achieve by increasing the temperature from 25 °C to 60 °C.	The mechanical properties of the printed part have a negative effect when the temperature crosses the glass transition temperature of the printed part. The separation force increases beyond the temperature of 35 °C.

## Data Availability

Not applicable.

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
