# Peer review of "A Review of Critical Issues in High-Speed Vat Photopolymerization"

_polymers, 2023, doi:10.3390/polym15122716_

Round 1

Reviewer 1 Report

This paper presents an extensive review of state-of-the-art in the field of high-speed projection vat photopolymerization, which is of great interest at the present time for the manufacturing community. In particular, the review focuses on the proposed solutions to increase the printing speed by decreasing the separation force to detach the cured part from the vat, or by decreasing the time needed for the resin refill. Moreover, the review describes the proposed solution to increase the performance of the projection system. Eventually, the paper identifies some meaningful future research directions that might be explored.

This review is a useful resource to learn about the main factors affecting the performance of projection vat photopolymerization, but its quality and readability could be improved (see below for detailed comments).

DETAILED REVIEW

Title

-     Specify that the review focuses on the projection vat photopolymerization.

1. Introduction

-   pag 1, lines 44-45: VPP cannot be compared to all other AM technologies, but only to the other technologies that used for the same materials used in VPP (mainly polymers, but also ceramics).

-     pag 1, lines 45-47: In a first classification TPP can be distinguished from SLA and projection VPP, which are based on one-photon polymerization.

-     pag 1, lines 46-47: SLA stands for “stereolithography” and not for “stereo lithography appereance”.

-     pag 2, lines 48-49: Another kind of dynamic mask used in projection VPP is the liquid crystal on silicon (LcoS) chip [for example, see: Zheng, X., Deotte, J., Alonso, M. P., Farquar, G. R., Weisgraber, T. H., Gemberling, S., ... & Spadaccini, C. M. (2012). Design and optimization of a light-emitting diode projection micro-stereolithography three-dimensional manufacturing system. Review of Scientific Instruments, 83(12), 125001].

-     pag 2, lines 53-55: The recoating with fresh resin is not always carried out, e.g. it is not carried out in systems with a “bat”/“bottom-up”/“constrained surface” configuration.

-     pag 2, lines 56-58: The two-photon photopolymerization is not necessarily performed using two laser beams, for example, the Nanoscribe system uses only one laser beam.

-     pag 3, lines 79-81: This clarification is not very relevant to the review topic and should be explained in more detail.

2. High-speed vat photopolymerization

-     pag 4, lines 121-124: This should be stated at the paragraph beginning, since the printing speed is not the only constrain to the productivity of the AM processes.

-     pag 4, line 113: “[…] which depends on the applications and can’t cannot change arbitrarily.”

3. Separation Force

-     pag 5, lines 134-135: The separation of the cured part from the vat surface is an issue only in the systems with a “constrained surface” configuration, therefore, in Section you have also to explain the classification of VPP systems according to their configuration (“bat”/“bottom-up”/“constrained surface” configuration vs. “bath”/“top-down”/“free surface” configuration).

-     pag 5, line 135: “[…] main challenges to increasing increase the printing speed […]

-     pag 5, line 137: “[…] it’s it is also important […]

-     pag 5, lines 138-139: “[…] accuracy of the printing printed part, […]

-     pag 5, lines 138-140: You should explain why the printable size and printing speed are affected by the separation force.

3.1. Separation Mechanisms

-     You could avoid reporting the equations and some calculation details of the different models, because they are slightly out of the main focus of this review and the reader can find them in the cited papers. [l]

3.2. Interface Modifications

-     pag 8, lines 250-252: This sentence is unclear and has to be fixed.

-     pag 8, lines 252-254: Explain which curve is being considered.

-     pag 10, lines 310-311: “Pan et al. [41], studied the effect of the printing area-to-perimeter (A/L) ratio has been studied by using the PDMS interface.”

-     The whole section is rather confused and hard to read, and it is difficult to understand where the descriptions of the different papers and proposed solutions begin and end. To make the section clearer, you could make a different paragraph for each study or for studies proposing similar solutions, and state the authors' names at the description beginning. In addition, you might consider adding a synoptic table that classifies the papers according to considered technology (DLP, LCD, etc.) and categories of proposed solutions. [l]

3.3. Process modifications

-     pag 13, lines 395-397: Wu et al. [70], derived an analytical model has been derived based on bilinear CZM to understand the effect of tilting motion on the cohesive stiffness and separation energy.

-     pag 13, lines 434-435: Ozkan et al. [73,74], carried out a hot lithography VPP process has been carried out with two different viscous polymer resins at different temperatures (25℃, 35℃, 45℃ and 55℃).”

-     Same comment as l.

4. Photopolymer resin refilling

-     pag 16, lines 466-467: He et al. [80], presented an interface modification approach has been presented by texturing the PDMS-coated interface layer.”

-     pag 16, lines 473-474: This sentence is unclear and has to be fixed.

-     Same comment as l.

-     Same comment as l.

5. Challenges with high-speed VPP image projection systems

-     Same comment as l.

6. Concluding remarks and future works

-     pag 25-26, lines 767-784: You should expand the discussion on the topics to be explored in future studies.

The overall writing quality is adequate, but could be further improved.

Author Response

Thanks for your detailed comments. The modifications have been done according to the comments. Please see the attached files for details responses.

Reviewer 2 Report

The paper includes a detailed review of available VPP AM technologies. Based on the title, there should be some critical descriptions, but I can see only some specific details about the technologies. Also English needs to be improved - you cannot use such form as can't (line 224) - it is a nonformal language that should not be used in academic language.  

I listed all my specific comments below: 

1. Please carefully read the instructions for authors in the case of abstract part. It needs to be rewritten because it has some very general descriptions of VPP technologies that should be a part of the introduction. 

2. If you provided the abbreviation description you do not have to do it once again. Please provide the description once and then continue the manuscript only with an abbreviation.

3. Please avoid very long captions in your figures (i.e. figure 5) provide only some essential descriptions and the rest move into the manuscript. 

4. I cannot see any significant criticism throughout the whole manuscript. The authors need to list all important issues of the mentioned technologies and review some available research results. In the present form, it looks more like a detailed specification of the VPP technologies. 

5. I think the authors need to put some data about postprocessing related to UV exposure after printing because it is a crucial part of some VPP technologies that increases the total time of production of the parts ready for application. 

I put my comments in the comments and suggestions for authors already. 

Author Response

Thanks for your comments and suggestions. The modifications have been done based on your comments. Please check the attached file for detail responses.

Round 2

Reviewer 1 Report

See the attachment.

Author Response

Thanks for your comments. All the comments are appropriately responded. Please have a look at the file I've attached.

Reviewer 2 Report

The authors made all-sufficient corrections. The paper could be published.  

Author Response

Thanks for your approval.